# LINE-1 transposition into murine *Thyroglobulin* results in congenital thyroid dysplasia

**Wendy J. Bailey**[1,‡], **Bart M. G. Smits**[2,‡,*], **Zoltan Erdos**[1], **John M. Gaspar**[1],
**Pamela Lane**[1], **Sabu Kuruvilla**[1], **Thomas W. Rosahl**[1], **Douglas Thudium**[1], **Jingzhou Wang**[1],
**Warren E. Glaab**[1], **Melissa MacGowan**[2], **Heather Multari**[2], **Christine Cumo**[2], **Adam Navis**[2],
**Thomas Forest**[1]

**1** Merck & Co., Inc., Rahway, New Jersey, United States of America, **2** Taconic Biosciences, Rensselaer, New York, United States of America

☯ These authors contributed equally to this work.
‡ These authors share co-first authors on this work.
* bart.smits@taconic.com

## Abstract

A spontaneous mutation in the wild type C57BL/6NTac mouse was discovered that is associated with early-onset histopathologic sequalae typical of thyroid dysplasia. The spontaneous mutation resulted from insertion of a L1 long interspersed nuclear element (LINE-1) into an intron within the Thyroglobulin (*Tg*) gene. The mouse genome contains a significant amount of retrotransposon DNA, and these mobile genetic elements routinely change genomic location through retrotransposition, including in germ cells. Analysis of the thyroid transcriptome suggested that the presence of the LINE-1 interferes with the *Tg* gene splicing, resulting in exclusion of exon 26 from most *Tg* transcripts in animals homozygous (HOM) for the insertion. The LINE-1 insertion allele of the *Tg* gene has been designated $Tg^{tdys\text{-}Tac}$. The resulting phenotype is inherited in an autosomal dominant manner with affected mice exhibiting thyroid follicular cell dysplasia that progresses to thyroid adenoma by 9 months of age with complete penetrance in homozygotes. Serum thyroid hormone measurements revealed a decrease in triiodothyronine (T3) levels in homozygotes at 12 months of age, as well as a decrease in tetraiodothyronine (T4) levels at 6–9 months and at 12 months of age in both heterozygotes and homozygotes. In addition, serum Thyroid Stimulating Hormone (TSH) level was strongly increased in homozygotes at 6–8 months of age, consistent with hypothyroidism. Computational molecular modeling showed that omission of the 64 amino acids from the TG protein arm domain, which is the consequence of exon 26-skipping in the *Tg* transcript, results in decreased local stability. This result in combination with the observed up-regulation in unfolded protein response (UPR) pathways in the thyroids of affected animals, identifies the arm domain of TG as important for its proper cellular distribution. This report describes a spontaneous retrotransposon insertion causatively linked to dysregulated physiological phenotypes in a widely used inbred mouse strain.

**Data availability statement:** All relevant data are within the manuscript and its Supporting Information files or submitted to the Gene Expression Omnibus.

**Funding:** The author(s) received no specific funding for this work.

**Competing interests:** The authors have declared that no competing interests exist.

## Introduction

The thyroid gland is part of the endocrine system responsible for regulating metabolism in mammals. The thyroid synthesizes and secretes thyroid hormones (TH), namely triiodothyronine (T3) and thyroxine (T4). The production of T3 and T4 is primarily regulated by a feedback loop starting with the hypothalamus secreting thyroid releasing hormone (TRH) this in turn, stimulates the pituitary gland to produce and release thyroid-stimulating hormone (TSH). TSH then signals the thyroid to releasing T3 and T4. These hormones are responsible for regulating heart rate, breathing, weight, body temperature and cholesterol levels among other functions. Thyroid follicles are the functional unit of the thyroid responsible for TH production and secretion from a spherical structure made up of an outer monolayer of thyrocytes and a central lumen filled with colloid consisting of highly concentrated TG. TG is a secretory protein synthesized in the thyrocyte endoplasmic reticulum (ER) [1]. This homodimeric glycoprotein functions include iodide storage and TH biosynthesis. Structurally, TG is made up of repeating domains designated region I, II-III, and the cholinesterase-like (ChEL) domain. TG maturation involves intracellular transport into the endoplasmic reticulum, where it starts to fold with the assistance of thyrocyte ER chaperones and oxidoreductases, as well as coordination of distinct regions of TG, to achieve a native conformation [2,3]. A functional attribute of the TG protein is the folding of regions I-II-III and ChEL into an independent functioning unit serving to stabilize the homodimerization of the ChEL domain which in turn serves as an intramolecular chaperone escorting TG through the secretory pathway [3]. Among vertebrates, an array of spontaneous mutations in TG have been described that cause congenital hypothyroidism and thyroidal ER stress [4–8]. These TG mutants are unable to achieve a native conformation within the ER, interfering with the efficiency of TG maturation and export to the thyroid follicle lumen for iodide storage and hormonogenesis [9].

LINE-1 are retrotransposons that make up 17% and 20% of the human and mouse genomes respectively [10,11]. Full-length LINE-1 can be autonomously active as they encode open reading frames ORF1 and ORF2 required for retrotransposition. LINE-1 inserted in the forward orientation within genes are largely inactivated due to probable deleterious effects on gene expression [12,13]. This is documented as the majority of the 500,000 LINE-1 copies found in mammalian genomes are truncated and/or rearranged. In humans, only about 0.02% remain full-length, retaining the ability to be transcribed and to transpose to a novel location. However, when retrotransposition happens in a germ cell, the novel allele may be passed on to the next generation and subsequently spread in the population.

Alterations of gene sequences that make up the canonical splice donor site (SD or 5' splice), or splice acceptor site (SA or 3' splice) may either lead to exon skipping, or it may result in the usage of cryptic splice sites in the vicinity [14]. While full-length LINE-1 capable of retrotransposition are rare, there are numerous examples of transposable elements disrupting normal gene expression in mice after translocating into non-coding regions including causing deafness due to a structural

defect in the cochlear stria vascularis [15], an obese phenotype by mutating the *ob* gene [16], an autoimmune phenotype by disrupting the *lpr* gene encoding the Fas antigen [17] and a complex motor disorder due to aberrant splicing of the *Glyrb* gene [18].

C57BL/6 is one of the most used mouse strains in biomedical research. C57BL/6 substrains maintained at different facilities acquire spontaneous mutations, which could have phenotypic consequences. In this report, we describe the discovery of an intronic LINE-1 in the *Tg* gene in the wildtype C57BL/6NTac (B6NTac) mouse maintained at Taconic Biosciences. This spontaneous mutation was shown to correlate to animals which exhibited thyroid dysfunction, as displayed by decreased serum TH levels, and thyroid follicular cell dysplasia that progressed to thyroid adenoma by 9 months.

## Results

### Thyroid dysplasia is associated with aberrantly spliced *Tg* transcripts

We discovered that animals from several independent mutant strains on the B6NTac background unexpectedly showed severe thyroid dysplasia, while several other mutant strains on the B6NTac background and wild type animals on the C57BL/6NCrl (B6NCrl) background were unaffected. We hypothesized that a spontaneous mutation could have arisen in the B6NTac colony affecting a portion of the strain.

To understand the molecular nature of the thyroid dysplasia (phenotype described in detail below), we focused on the *Tg* gene due to its essential role in thyroid function, its abundant expression in the thyroid tissue and the existence of characterized mutations resulting in thyroid dysfunction.

### RT-PCR analysis of the *Tg* transcript

We designed successively positioned sets of primers with an expected amplicon length of approximately 2 kb spanning regions from exons 1–9, 9–18, 17–31, 30–38, 38–48 of the *Tg* transcript (S1A Fig). Transcriptional analysis by RT-PCR resulted in amplicons of expected length in regions from Exon 1–17 and 31–48 for all affected and unaffected mice (S1B Fig). Initial amplification of the region from exon 17–31 was not successful in affected animals. Results showed multiple incorrect size bands for all affected animals. Amplification of shorter sections of this region was successful and showed a truncated amplicon containing exons 22–32 in affected mice only (S1B Fig).

Using RNA sequencing, we compared the entire thyroid transcriptome of 5 affected animals on the B6NTac background to 7 unaffected animals on the B6NCrl background (Fig 1A). Interestingly, an alternative splice form of the *Tg* gene was discovered in the affected mice, but not in the unaffected mice. The five affected mice all had most reads (73.4–76.3%) joining exon 25 with exon 27 indicating aberrant splicing that skips exon 26. The unaffected mice all had fewer than 0.25% of reads in this category (Fig 1B). Skipping of exon 26 results in an in-frame deletion of 64 amino acids from the TG protein product. These data suggest that the thyroid of affected mice predominantly produce the variant *Tg* transcript, while only a portion of the *Tg* transcripts are full-length.

### Skipping of exon 26 is caused by the presence of a LINE-1 in intron 25

To find the variant DNA sequence underlying the aberrant splicing event in the affected animals, we performed whole genome sequencing on genomic DNA from five affected and seven unaffected mice. Close examination of the read alignments just upstream of exon 26 of *Tg* revealed a cluster of genomic variants and decreased read coverage (Fig 2). There were few reads spanning this location, and those present had one end with several different variants. These reads were used to retrieve additional unaligned reads that provided more evidence for an additional sequence of a large insertion at this genomic locus. To find the identity of this putative insertion, the ends were aligned to the mouse reference genome via bowtie2 [19]. There were 3,722 valid alignments in the mouse genome, at distances of 6−7 kilobase (kb) from each other and overlapping the ends of LINE-1 sequences. Thus, the read evidence was indicating that the affected mice had a LINE-1 that had inserted into intron 25 of *Tg*, 15 base pairs upstream of the splice acceptor site of exon 26.

**A)**

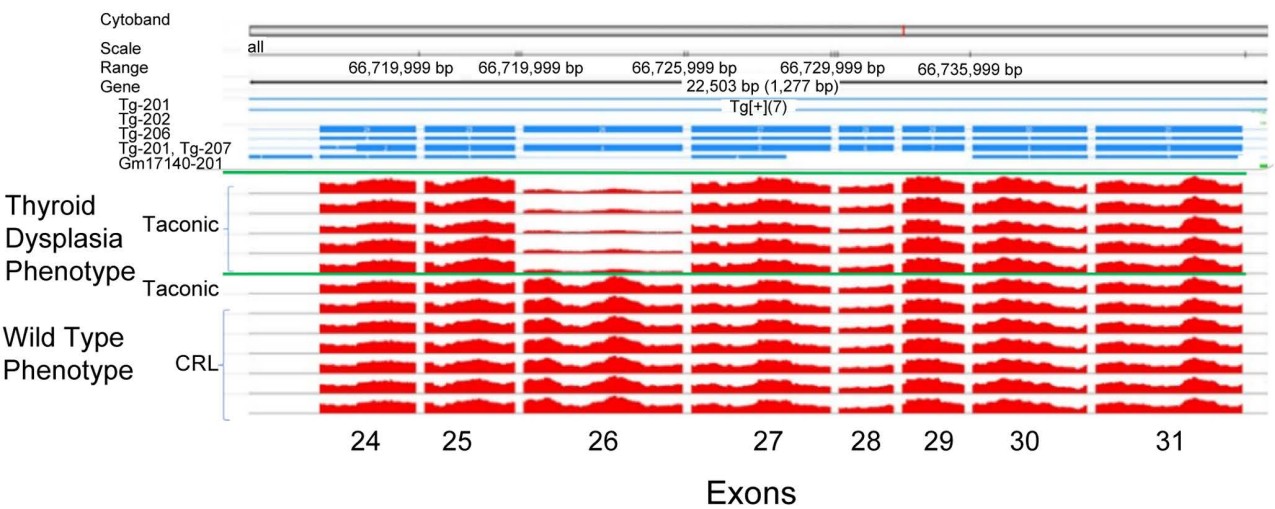

**B)**

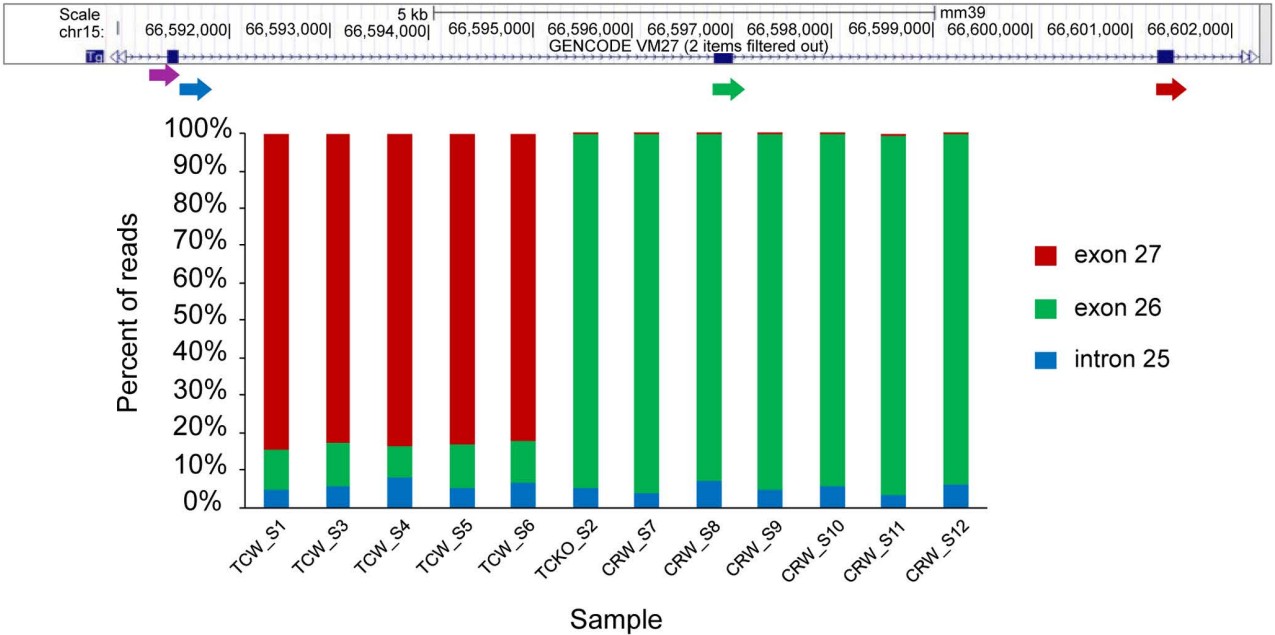

**Fig 1. RNA analysis reveals a~6-fold reduced coverage of exon 26 in the murine *Tg* gene.** (A) *Tg* transcriptional coverage in B6NCrl (CRL) and B6NTac (Taconic) thyroid samples determined by RNAseq. Shown are exons 24 through 31 of the murine *Tg* gene. Coverage of exon 26 is ~6x decreased in B6NTac affected versus unaffected B6NTac and B6Crl samples. (B) Percentage of RNAseq reads continuing downstream of *Tg* exon 25 (indicated by purple arrow). A custom python script was used to count the 10 nucleotides found downstream of the 3' end of exon 25. Three types of transcripts were observed continuing from the 3' end of exon25: 1) intron 25, indicating unspliced RNA (blue arrow), 2) exon 26, indicating normal splicing (green arrow) or, 3) exon 27, indicating aberrant splicing that skipped exon 26 (red arrow). The five affected B6NTac mice all had the majority of reads (73.4-76.3%) continuing with exon 27, whereas the unaffected mice all had less than 0.25% of reads in this category.

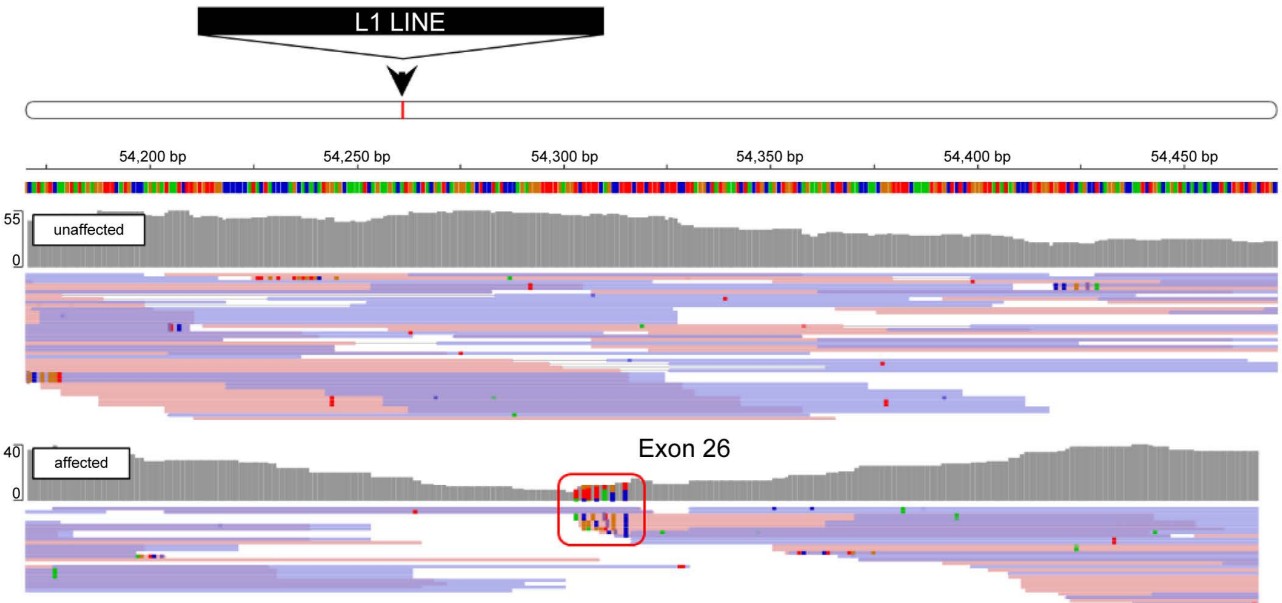

**Fig 2. Whole genome sequencing map of the intron 25-exon 26 transition of the murine *Tg* locus.** A variant cluster indicated by the red square is present in affected (lower track) but not unaffected (upper track) animals in intron 25, near the 5' end of exon 26. Variant cluster reads were used for the identification of a LINE-1 element insertion in the intron between exon 25 and 26.

In addition, long read (nanopore) sequencing was used to read through the insertion. Reads encompassing the junction between the LINE-1 and the *Tg* intron were collected and assembled into a consensus sequence [20]. A total of 56 reads were included in the assembly. The full-length insertion sequence is estimated to be 6.7 kb, consistent with the size of a full-length LINE-1. The sequence data indicate that the LINE-1 has integrated into intron 25 with the ORF1 and ORF2 sequences in antisense orientation with respect to the *Tg* coding sequence (S1 File).

### Thyroid dysplasia is inherited in an autosomal dominant fashion and is strongly associated with a LINE-1

Based on the genomic sequence data we developed a genotyping assay capable of distinguishing wild-type (WT), heterozygous (HET) and homozygous (HOM). The assay was used to genotype our pedigreed colony (Fig 3A). A heterozygous breeding pair produced 13 pups of which 3 were HOM, 4 HET and 6 WT. All 13 pups and parents were evaluated for thyroid pathology by histological analysis when the pups reached 10 weeks of age. All 3 HOM and all 4 HET pups as well as both HET parents showed thyroid dysplasia and the WT pups did not, indicating that the phenotype is strongly associated with the LINE-1 in *Tg* and inherited in a dominant fashion. Additional genotyping and phenotyping of the animal cohorts produced for this study indicated that there is complete penetrance of the thyroid dysplasia phenotype in animals with the LINE-1. Presence of the LINE-1 was found in frozen stock from the B6Ntac colony established in 2014, suggesting that the actual retrotransposition event occurred before that time.

Using RT-PCR we then determined exon 26-included *Tg* transcripts as a proportion of total *Tg* transcripts produced in the thyroids of WT, HET and HOM animals. The level of exon 26-included transcripts is determined as the average of the level of exon 25–26 junctions and exon 26–27 junctions. The level of total *Tg* transcripts is determined as the level of exon 24–25 junctions. The proportion of exon 26-included transcripts in WT was 1.049, in HET was 0.662 and in HOM was 0.245 (Fig 3B, S2 File), indicating that HETs and HOMs respectively produce 63% and 23% of the level of exon 26-included transcripts as compared to WT.

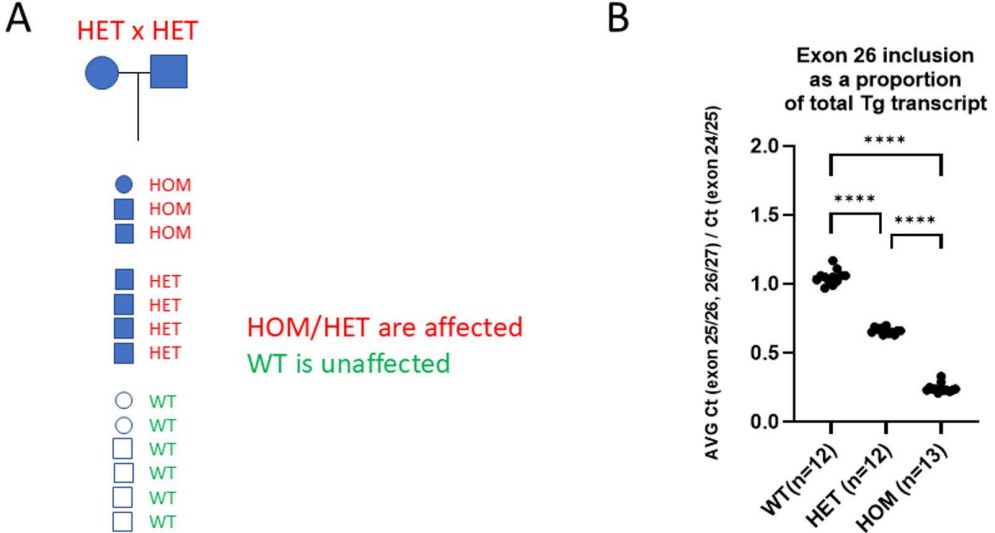

**Fig 3. Genotype-based analysis of the LINE-1.** (A) Pedigree analysis indicating that thyroid dysplasia is strongly associated with the LINE-1 and is inherited in a dominant fashion. Affected animals indicated in red font and filled symbols, unaffected animals indicated in green font and open symbols. (B) Exon 26 inclusion in the *Tg* transcript in WT, HET and HOM thyroid samples. Graph plots the ratio of the Ct value for the exon 25-26 and exon 26-27 junctions over the Ct value for the exon 24-25 junction, relative to WT. The exon 25-26 and 26-27 junctions are affected by the skipping of exon 26. The data were analyzed by One-way ANOVA with Bonferroni correction. Significance is indicated by **** at adjusted $p < 0.0001$.

## Thyroid dysplasia is congenital in heterozygotes and homozygotes, and progresses to thyroid adenoma in homozygotes at 9 months of age

We performed histological, immunohistochemical and ultrastructural analysis of the thyroid gland in unaffected B6NCrl, unaffected B6NTac mice, and affected B6NTac mice carrying the LINE-1 insertion.

In the mice with the LINE-1 insertion we observed thyroid dysplasia, which was characterized in H&E-stained histology slides in young adult mice (12 weeks of age) by distension of the perinuclear cytoplasm of the thyroid follicular epithelium with a sharply delineated basolateral amphophilic accumulation, decreased follicular size, less eosinophilic staining of follicular colloid, and occasional exfoliation of degenerate thyroid follicular epithelial cells into the follicular space (Fig 4A–C). Mice HET for the LINE-1 insertion exhibited a less severe phenotype than mice HOM for the insertion (Fig 4A–C). Dysplasia was absent in thyroid of WT mice.

Immunohistochemical staining with an anti-Thyroglobulin antibody demonstrated that the basolateral amphophilic accumulation in affected (B6NTac) mice contained TG, and that the follicular colloid stained less intensely for TG than unaffected (B6NCrl) mice (Fig 4D–E). Transmission electron microscopy (TEM) demonstrated that the basolateral accumulation in affected mice was present within the endoplasmic reticulum (ER), whereas unaffected mice showed normal ER morphology (Fig 4F–G).

In adenohypophysis of the pituitary of affected mice there was an increase in large, lightly basophilic cells at 12 months of age (Fig 4H–I) which stained by immunohistochemistry with an anti-thyroid stimulating hormone antibody for TSH (Fig 4J–K). Moreover, the intensity of the IHC staining for TSH in these large pituitary cells in affected mice was generally less intense than the TSH staining in pituitary cells in unaffected mice.

Older affected mice (> 6 months of age) exhibited proliferative changes in thyroid follicular epithelial cells, characterized by clusters of basophilic cuboidal cells that commonly formed a double layer around thyroid follicles, as well as an increase in mixed inflammatory cell infiltrates in the thyroid interstitial space. With additional time (at and beyond 9 months of age) thyroid follicular adenomas were observed in some mice (Fig 4L–M).

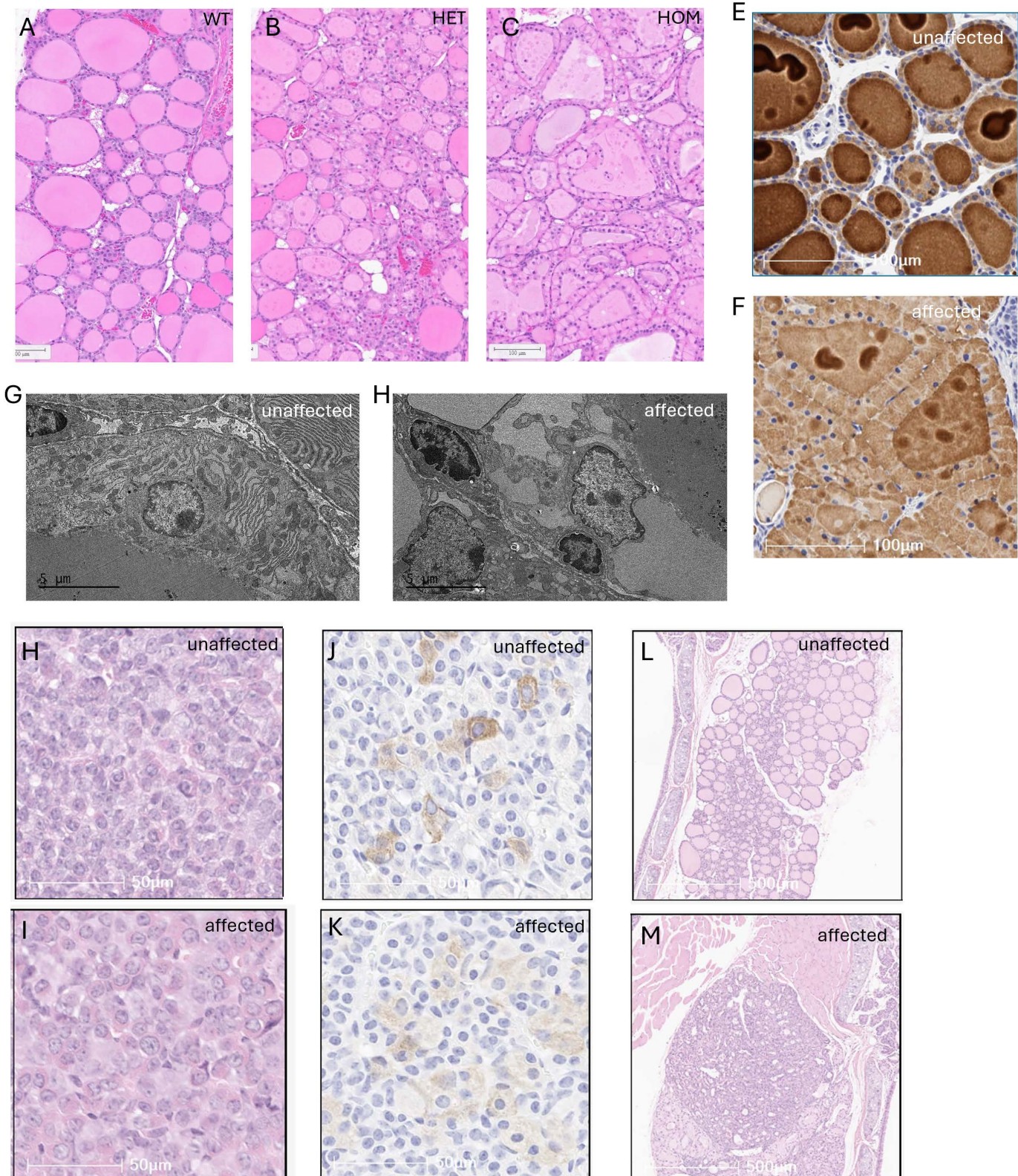

**Fig 4. Thyroid dysplasia in inbred C57BL/6NTac mice WT, HET or HOM for the LINE-1.** (A-C) Representative light microscopic morphology of the thyroid follicular epithelial cells of B6NTac WT, HET and HOM at 12 wks of age. Thyroid dysplasia most developed in the HOM animals and to a lesser

degree in HET animals. (D-E) Immunohistochemistry staining of thyroid follicular epithelial cells from B6NCrl unaffected and B6NTac affected animals using anti-Thyroglobulin antibody. Accumulation of TG is visible in the cytoplasm of thyroid follicular epithelial cells and the associated decrease in TG staining in the follicular colloid of affected mice (B6NTac) but not in unaffected mice (B6NCrl). (F-G) Representative transmission electron microscopy images of thyroid follicular epithelial cells. The images illustrate the ultrastructural dilation of the endoplasmic reticulum in B6NTac affected mice and not in B6NCrl unaffected mice. (H-K) Representative light-microscopic images of the adenohypophysis of the pituitary in B6NTac affected and B6NCrl unaffected mice. The images from H&E stained tissues (H-I) illustrate an increase in large lightly basophilic cells in the adenohypophysis of the pituitary in affected mice as compared with unaffected mice. The immunohistochemistry staining (J-K) using anti-Thyroid Stimulating Hormone (TSH) shows that the large lightly basophilic cells stain lightly for TSH. (L-M) Representative light microscopic morphology of the thyroid follicular epithelial cells of B6NCrl unaffected and B6NTac affected mice at 12 months of age. The images illustrate the presence of proliferative, inflammatory, and neoplastic changes in older affected mice, but not unaffected mice.

Once we discovered the LINE-1, we produced cohorts of WT, HET, and HOM to evaluate the onset and progression of the phenotype. At 3–4 weeks of age (HOM n = 11, HET n = 9, WT n = 10), 5–6 weeks of age (HOM n = 9, HET n = 11, WT n = 10) and 11–12 weeks of age (HOM n = 8, HET n = 16, WT n = 9), all homozygotes and heterozygotes showed thyroid dysplasia and all wild types showed normal thyroid histology. At 9 months of age (HOM n = 23, HET n = 15, WT n = 23), 12 months of age (HOM n = 9, HET n = 19, WT n = 10), and 18 months of age (HOM n = 20, HET n = 6, WT n = 7), all homozygotes progressed to thyroid adenoma, while all heterozygotes still showed thyroid dysplasia and the wild types showed normal thyroid histology (S2 Fig). No cases of adenocarcinoma were found in aged homozygotes.

These data indicate that the onset of thyroid dysplasia is before weaning and that the phenotype is likely congenital, in accordance with previously published rodent mutations in *Tg* [4,21]. The phenotype only shows progression to adenoma in homozygotes at 9 months of age, but not in heterozygotes up to 18 months of age.

### The LINE-1 in *Tg* does not affect body weight

As THs are well-known regulators of growth and body weight, we measured weight of animals 4 weeks of age (Female HOM n = 10, HET n = 9, WT n = 5), 6 weeks of age (Female HOM n = 9, HET n = 11, WT n = 10), 9 months of age (Male HOM n = 9, HET n = 4, WT n = 5; Female HOM n = 11, HET n = 7, WT n = 5), and 12 months of age (Male HOM n = 9, HET n = 19, WT n = 10). None of the age groups showed a difference in body weight between any of the genotypes in male or female (adj. p > 0.05), except for females at 12 weeks of age, which showed a small but significant increase in body weight in HOM (adj. p = 0.04) compared to WT (S3 Fig).

### Serum T3 and T4 levels are affected by the LINE-1 at 9 and 12 months of age

Since TG is the precursor to T3 and T4, serum T3 and T4 levels were measured for WT, HET and HOM animals at 4 weeks (T3 HOM n = 8, HET n = 9, WT n = 8; T4 HOM n = 9, HET n = 9, WT n = 8), 6 weeks (T3 HOM n = 7, HET n = 10, WT n = 8; T4 HOM n = 7, HET n = 11, WT n = 8), 9 months (T3 HOM n = 17, HET n = 10, WT n = 8; T4 HOM n = 19, HET n = 11, WT n = 9) and 12 months of age (T3 HOM n = 7, HET n = 18, WT n = 9; T4 HOM n = 8, HET n = 19, WT n = 10). T3 levels were not different between any of the genotype groups at 4 or 6 weeks or 9 months of age but were significantly reduced in HOM animals compared to WT (p = 0.04), and a trend to be reduced in HOM compared to HET (adj. p = 0.0501) animals at 12 months of age (Fig 5A–D). Similarly, T4 levels were not different between any of the genotype groups at 4 or 6 weeks or 9 months of age (Fig 5E–G). At 12 months of age both HOM (adj. p = 0.008) and HET (adj. p = 0.0001) animals show significantly reduced T4 levels compared to WT (Fig 5H). HOM also shows a trend towards reduced T4 levels as compared with HET (adj. p = 0.09) at 12 months of age (Fig 5H) indicative of a gene dosage effect on T4 levels at 12 months of age. TSH levels are shown for animals of 6–9mo of age (Fig 5I) with matching T4 levels for the same animals (Fig 5J). TSH levels are significantly increased in HOM animals compared to HET or WT animals (adj. p < 0.001). TSH levels are not different between WT and HET animals. The matching serum T4 levels for the same animals show a significant

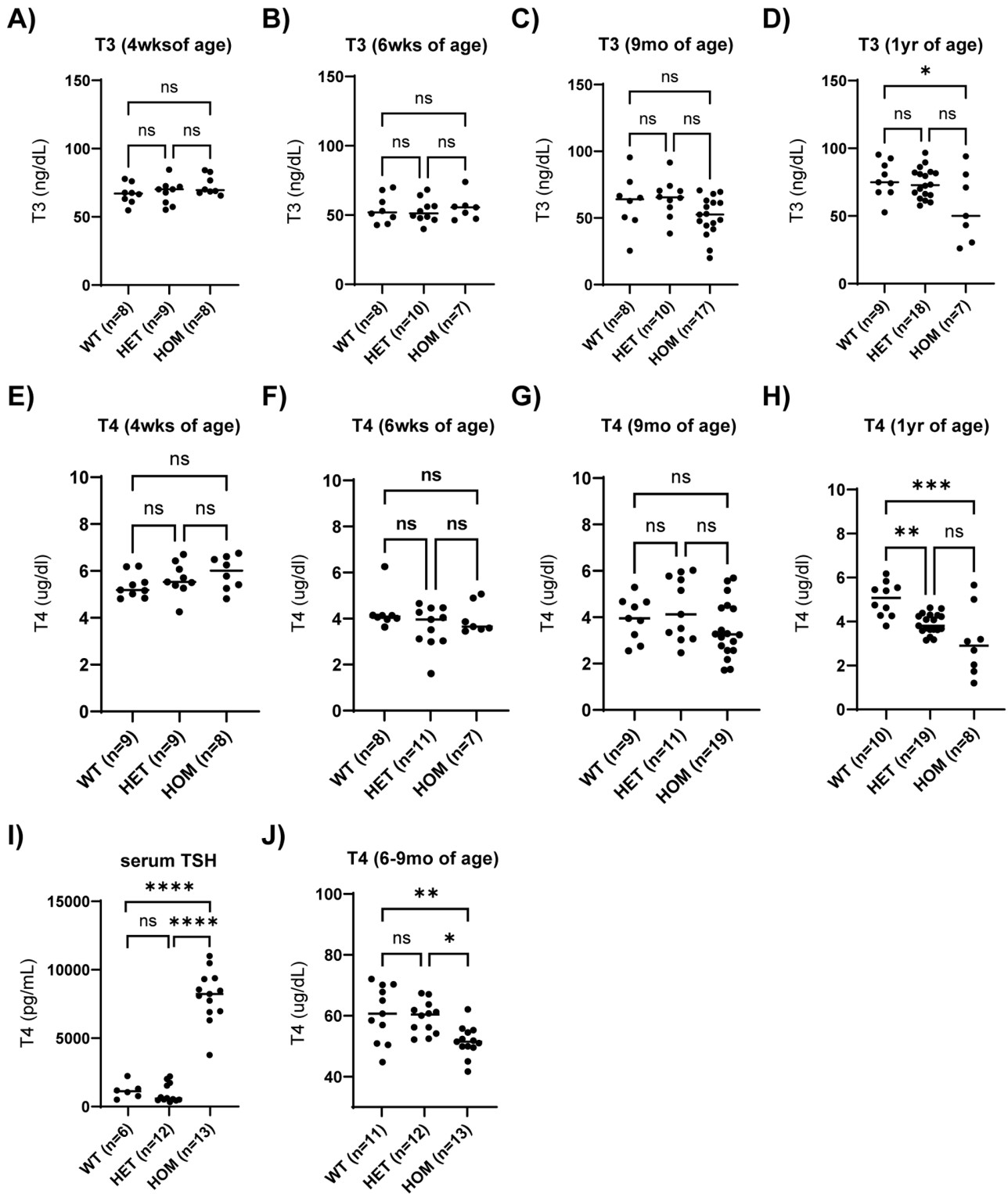

**Fig 5. Serum hormone levels in WT, HET and HOM animals.** (A-D) T3 levels are shown for animals at 4 wks (A), 6 wks (B), 9 mo (C) and 1 yr (D) of age. (E-H) T4 levels are shown for animals at 4 wks (E), 6 wks (F), 9mo (G) and 1 yr (H) of age. (I) TSH levels are shown for animals of 6-9mo of age. (J) Matching T4 levels for the same animals as in (I). All panels show individual measurements with the average indicated by a horizontal line. The data were analyzed by One-way ANOVA with Bonferroni correction. Significance is indicated with * at adjusted $p < 0.05$, ** at adjusted $p < 0.01$, *** at adjusted $p < 0.001$, and **** at adjusted $p < 0.0001$.

**Table 1. Ingenuity pathway analysis.**

| Symbol | Ensembl | Gene name | Expression fold change | Expression p-value | Expression False Discovery rate | Expression Intensity | Expected regulation direction |
|--------|---------|-----------|------------------------|--------------------|---------------------------------|----------------------|-------------------------------|
| 12 genes associated with unfolded protein response pathway | | | | | | | |
| CALR | ENS-MUSG00000003814 | Calreticulin | −3.3 | 1.4E-18 | 5.0E-16 | 24721 | UP |
| DDIT3 | ENS-MUSG00000025408* | DNA-damage inducible transcript 3 | −3.6 | 1.1E-19 | 4.2E-17 | 770 | UP |
| DNAJB9 | ENS-MUSG00000014905 | DnaJ heat shock protein family (Hsp40) member B9 | −2.3 | 2.4E-12 | 3.1E-10 | 2847 | − |
| DNAJC3 | ENS-MUSG00000022136 | DnaJ heat shock protein family (Hsp40) member C3 | −2.2 | 2.1E-12 | 2.7E-10 | 8002 | DOWN |
| EDEM1 | ENS-MUSG00000030104 | ER degradation enhancer, mannosidase alpha-like 1 | −1.8 | 5.5E-06 | 1.0E-04 | 2220 | UP |
| HSP90B1 | ENS-MUSG00000020048 | Heat shock protein 90, beta (Grp94), member 1 | −4.1 | 1.0E-27 | 1.3E-24 | 51385 | UP |
| HSPA5 | ENS-MUSG00000026864 | Heat shock protein 5 | −4.2 | 7.3E-25 | 5.5E-22 | 32386 | UP |
| HSPA9 | ENS-MUSG00000024359 | Heat shock protein 9 | −1.7 | 2.2E-09 | 1.5E-07 | 6103 | UP |
| P4HB | ENS-MUSG00000025130 | Prolyl 4-hydroxylase, beta polypeptide | −2.0 | 6.5E-11 | 6.3E-09 | 12480 | UP |
| PDIA6 | ENS-MUSG00000020571 | Protein disulfide isomerase associated 6 | −3.7 | 1.9E-24 | 1.3E-21 | 8188 | UP |
| SEL1L | ENS-MUSG00000020964 | Sel-1 suppressor of lin-12-like (C. elegans) | −3.2 | 1.5E-27 | 1.6E-24 | 9894 | UP |
| SYVN1 | ENS-MUSG00000024807 | Synovial apoptosis inhibitor 1, synoviolin | −1.9 | 4.2E-06 | 1.0E-04 | 1576 | UP |

reduction in HOM animals compared to WT and HET animals (adj. p = 0.006, adj.p = 0.01, respectively), but not in WT compared to HET animals.

### Molecular modeling of variant TG protein indicates that the omission of 64 amino acids impacts stability

To understand how variant TG can impact thyroid function, we performed molecular pathway analysis on the RNA sequencing data set. The Ingenuity Pathway Analyses application indicated that several pathways associated with unfolded protein response were up regulated in affected (B6NTac) versus unaffected (B6NCrl) mice (Table 1). Upregulation of such pathways would be consistent with protein misfolding.

As TG is the most abundant protein in thyroid cells, we next explored the impact of the omission of 64 amino acids caused by exon 26-skipping on TG protein structure by in silico molecular modeling. The "Arm" fragment of mouse TG with or without the 64 amino acid omission was respectively modeled by AlphaFold 2 (AF2) [22–24]. WT "Arm" of mouse TG was predicted with high confidence (average pLDDT = 81) and closely resembled the experimentally determined structure of human TG "Arm" (Fig 6). Remarkably, the peptide encoded by exon 26 appeared as a well-modeled (average pLDDT = 86) and structured core flanked by long peptide strands with low pLDDT at both N and C termini (Fig 6A–B). Upon omission of the 64 amino acids, a significant drop in pLDDT was observed in the above-mentioned peptide strands (Fig 6B), suggesting that the mutation promotes local unfolding, as AF2 predictions with low pLDDT score are frequently associated with flexible and disordered structure [ 23,25,26]. Further analysis of the modeled structures revealed that with

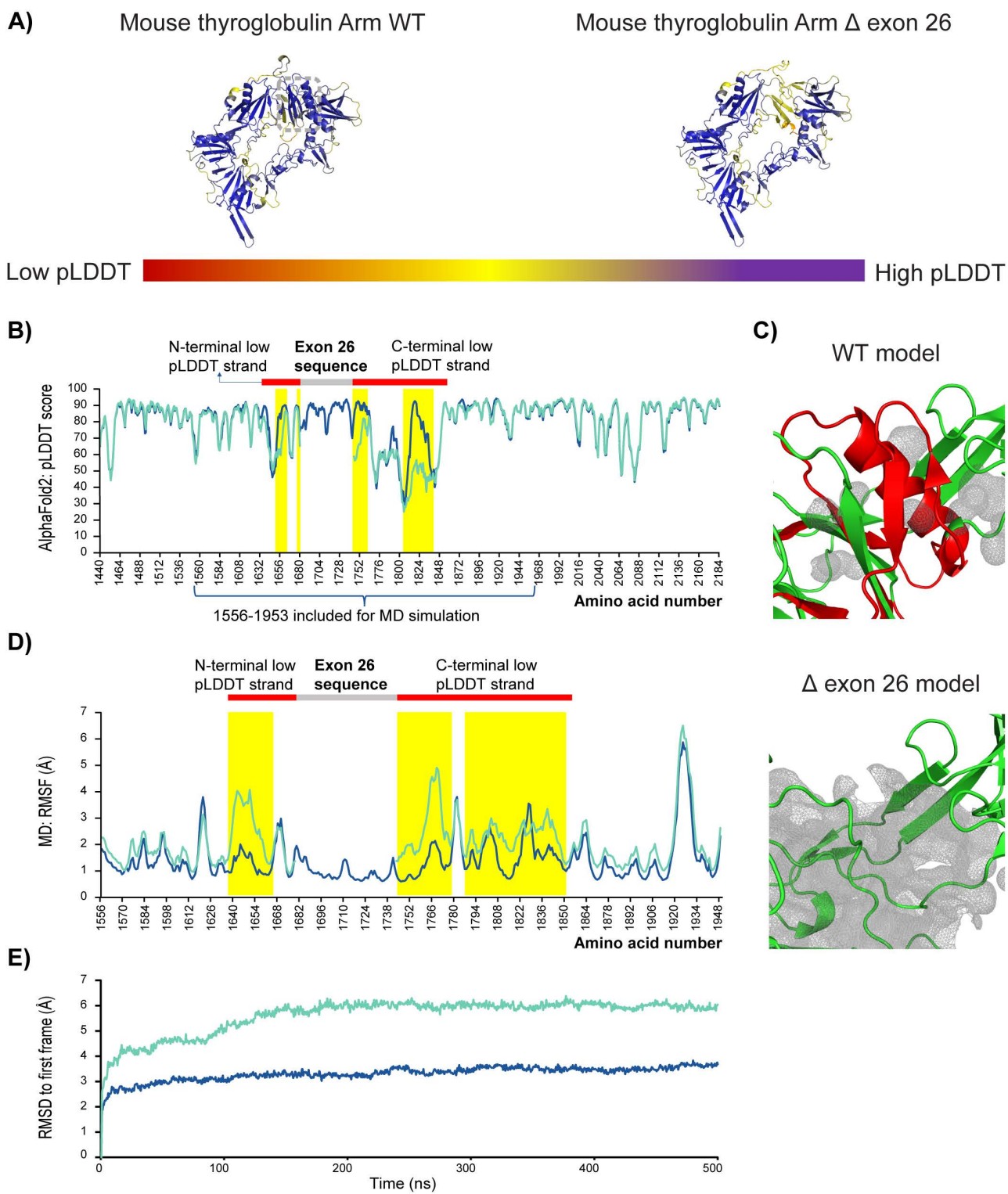

**Fig 6. TG protein modeling.** (A) AlphaFold2 predicted Arm segment of mouse TG with or without the 64 amino acid omission due to exon 26 skipping is shown, with each residue colored by its pLDDT score from structure prediction. High pLDDT residues are colored as blue, and low pLDDT residues are colored as red. The position of exon 26 encoded sequence is boxed in green dash line. (B) Position-specific pLDDT comparison of AlphaFold 2

predicted mouse TG Arm with (red line) or without (blue line) 64 amino acid omission. Positions of exon 26 encoded region, extended stretch of low pLDDT segments towards the N (1638-1680) and C (1745-1856) termini of exon 26 encoded sequence, and regions with significant drop in pLDDT upon exon 26 skipping (yellow bars) are respectively high-lighted. (C) Cavity analysis of WT and exon 26 skipped structural models. Exon 26 encoded sequence is colored red, and cavities near its position in each structure are shown as grey meshed surfaces. (D) Root-mean squared fluctuation (RMSF) analysis of molecular dynamics simulation outputs for WT (blue line) and exon 26 skipped (red line) mouse TG structures initially modeled by Alpha-Fold2. Regions with significant increase in fluctuation upon exon 26 skipping are highlighted in yellow bars. Exon 26 sequence and two long stretches of segments with low pLDDT scores (highlighted in (C)) on its N and C termini are also annotated for reference. (E) Root-mean squared deviation of MD plotted for WT (blue line) and exon 26 skipped mouse TG structures.

the omission of the 64 amino acids, the tight core packing mediated by the encoded peptide was replaced by large cavities (Fig 6C), indicating disrupted interaction network within the structure.

Molecular dynamics (MD) simulation was conducted to directly probe the physical effect of the mutations on the Arm fragment. Root-mean squared fluctuation (RMSF) of each amino acid position during the simulation was compared to reveal position-specific dynamic flexibility. The average RMSFs for the wild type and mutant sequences were respectively 1.4 Å and 2.1 Å. The difference was contributed by local RMSF increase of 2–3 Å in multiple regions upon the mutation, especially among regions with low pLDDT scores in AF2 prediction and close to the removed sequence, supporting the destabilization effect of the mutation (Fig 6D). Remarkably, the low pLDDT regions in the AF2 models in general demonstrated higher flexibility than other regions included in the MD simulations. The exon 26-encoded peptide was more stable (average RMSF = 1 Å) than its neighboring sequence during the simulation, supporting its role as a fold-stabilizing core, and coinciding with its high pLDDT scores in AF2 modeled structure. Despite regional changes of pLDDT and RMSF measurements, most other components of the modeled structures in both AF2 and MD simulations (Fig 6B–D) returned similar pLDDT and RMSF values upon omission of the 64 amino acids, suggesting the overall fold could still be realized, but likely at lower stability due to the more significant presence of disordered components (Fig 6D, S4 Fig). Plotting the root-mean squared deviation (RMSD) of MD snapshots against the corresponding first frame also revealed a larger conformational shift for the mutant as time progressed (Fig 6E), further indicating the compromised stability due to exon 26 skipping.

## Discussion

Animals on the C57BL/6NTac background exhibited thyroid dysfunction associated with insertion of a LINE-1 retrotransposon into the intron upstream of exon 26 in the *Tg* gene, resulting in variable degrees of exon 26 skipping in the *Tg* transcript. Molecular pathway analysis indicates activation of an unfolded protein response and transmission electron microscopy assessment shows an accumulation of misfolded TG in the ER of thyroid follicular epithelial cells of affected animals. The histomorphological and histochemical findings include thyroid follicular cell dysplasia that progressed to thyroid adenoma by 9 months, hypertrophy and hyperplasia of TSH secreting cells in the pituitary, decreased serum thyroid hormone, and increased TSH. The LINE-1 insertion had minimal impact on the overall phenotype with no evident changes in body weight. We named the spontaneous mutation caused by LINE-1 insertion into the *Tg* gene, *Tg^{tdys-Tac}*.

Our observations are supported by the well understood hypothalamic–pituitary–thyroid axis responsible for the regulation of metabolism [27]. The hypothalamus responds to low T3 and T4 levels by releasing thyrotropin-releasing hormone (TRH), which in turn, stimulates the thyroid to generate more T3 and T4 until normal blood levels are reached. In a classic feedback mechanism, THs signal the hypothalamus to down-regulate the release of TRH and to reduce TSH release from the pituitary gland to maintain metabolic homeostasis. Our study demonstrated a decrease in serum T3 and T4 due to a spontaneous mutation, which stimulated an increase in TSH thereby driving proliferation of the thyroid epithelial cells, ultimately leading to hyperplasia and eventually adenoma. In the mice evaluated, thyroid follicular cell adenoma was only observed in mice homozygous for the LINE-1 retrotransposon insertion but not in the heterozygous mice, even up to 18

months of age. We attributed this difference in incidence of adenoma to increased exon 26 skipping in homozygotes leading to decreased full-length *Tg* transcript level. We propose that the gene dosage effect of a greater decrease in full-length *Tg* transcript level would lead to a greater decrease in THs, a concomitant increased TSH release from the pituitary and therefore increased stimulation of thyroid follicular cells to proliferate in homozygotes compared to heterozygotes. This is supported by the observation that heterozygotes have a smaller decrease in serum THs and do not have the increased serum TSH level observed in homozygotes.

There are several rodent models for congenital hypothyroidism associated with *Tg* mutations causing defective TG folding and accumulation in the ER. The *cog* mouse model carries the L2263P mutation and displays autosomal recessive congenital hypothyroidism with adult goiter [4]. The WIC-rdw rat model on the other hand carries the G2298R mutation and displays congenital hypothyroidism without goiter [21]. When the G2298P mutation was introduced on the murine C57BL/6J background, the model displayed congenital hypothyroidism with goiter, suggesting that *Tg* mutations can have different disease outcomes because of genetic background. In both models, ER stress and thyroid cell death is severe. Both models show strongly reduced circulating levels of T3 and T4, accompanied by an increased circulating TSH level, which are the classical diagnostics of hypothyroidism.

There are several key differences between the *Tg^{tdys-Tac}* model and the *cog* and WIC-rdw model. First, our model does not show dwarfism. Our body weight measurements indicate no significant differences between HET, HOM and WT at any age. Second, serum T3 and T4 levels are not impacted at early stages of life in our model. We only observe significant differences in circulating T3 level at 9 months of age and circulating T3 and T4 levels at 12 months of age. At those ages thyroid adenomas are present at 100% penetrance in homozygotes, but not heterozygotes. Third, we did not observe goiters in heterozygous or homozygous *Tg^{tdys-Tac}* mice up to 12 months of age. It was found in the *cog* and WIC-rdw models that goitrous growth requires sustained increased thyroid cell proliferation [28], which may not be present or not be severe or sustained enough to support goitrous growth in our model. Interestingly, while human heterozygous carriers of *TG* mutations do not show signs of disease, in animal models heterozygotes do show thyroid phenotypes. Zhang et al. reported that heterozygous carriers of the *cog* mutation show signs of significant thyrocyte ER stress and individual thyrocyte cell death, which do not seem to affect the integrity of the surrounding thyroid epithelium [29]. Similarly, heterozygotes for the *Tg^{tdys-Tac}* mutation show thyroid dysplasia and signs of ER stress. It is possible that thyroids of human heterozygous *TG* mutation carriers show ER stress and follicular cell dysplasia without clinical signs of disease.

Targeted mutant *Tg* mice were generated by the Knockout Mouse Phenotyping Consortium (KOMP) at the Jackson Laboratory using the CRISPR/Cas9 technology (043760 - Strain Details (jax.org); Tg<em1(IMPC)J> Endonuclease-mediated Allele Detail MGI Mouse (MGI:6149762) (jax.org)). The alteration resulted in the deletion of 372 bp, including exon 3 and 274 bp of flanking intronic sequence, and is predicted to cause a change of amino acid sequence after residue 60 and early truncation 17 amino acids later. Phenotypic analyses of *Tg* mutant mice and C57BL6/NJ WT control mice were carried out as part of the International Mouse Phenotyping Consortium (IMPC). *Tg* mutant mice show – beside some relatively subtle behavioral and cardiac phenotypic changes – an improved glucose tolerance and increased fasting circulating glucose levels in comparison to WT controls (https://www.mousephenotype.org/data/genes/MGI:98733; https://phenome.jax.org/komp/genotypes/3199002?study=EAP). However, thyroid histology or testing of the thyroid function was not part of this IMPC phenotyping workflow. Future experiments should address that the introduced genetic mutation resulted indeed into a true null mutation and complete loss of TG function. Also, T3, T4 and TSH levels should be measured in the *Tg* mutant mice, as well as a search for potential compensatory mechanisms to explain those unexpectedly subtle phenotypic changes in the *Tg* mutant mice. Another murine presumably null (nonsense) *Tg* mutation was reported in a study describing a forward genetic ENU mutagenesis screen for recessive mutations affecting Heart Rate (HR) [30]. The existence of this mutation in homozygous state is another piece of evidence that *Tg* null or strong reduction of function mutations are viable.

LINE-1 retrotransposons are one of the most abundant classes of mobile DNA elements that account for approximately 20% of the human genome [11]. A full-length LINE-1 encodes a 5′-untranslated region (5′-UTR), two open reading frames

(ORF1 and ORF2) required for retrotransposition, and a 3′-UTR plus a poly-A tract. Recently a third ORF was discovered, namely ORF0, located on the opposite strand of the 5'-UTR [31].

Insertion of retrotransposons into introns of genes can impact gene function in several ways. For example, an intronic LINE-1 insertion can interfere with splicing and induce exon-skipping, an intronic LINE-1 insertion can result in exonization of the LINE-1 sequence and disrupt the endogenous transcript, a LINE-1 can cause exon inversion or exon deletion, and LINE-1 insertions can alter histone modifications affecting transcript levels [32–41]. In the Tg^{tdys-Tac} model, we show by RT-PCR analysis that skipping of exon 26 takes place when the LINE-1 is present. While the exact mechanism of exon-skipping would need to be investigated in future studies, we hypothesize that the location of the LINE-1 at a distance of 16 base pairs away from the exon 26 splice acceptor site interferes with proper loading of the splicing machinery on the pre-mRNA, but not in all cases. Although the splice acceptor site is still intact, the LINE-1's location with respect to the splice acceptor site suggests that the branch site sequence might be impacted, which is necessary for proper splicing. The branch site sequence in human splicing does not have a clear consensus sequence [42], but is typically located 20−50 base pairs upstream of the splice acceptor. We hypothesize that due to the LINE-1 insertion's effect on the branching in intron 25, exon 26 is skipped and branching happens preferentially in intron 26, splicing exon 25 to exon 27. An alternative hypothesis is that due to the LINE-1 insertion, a cryptic splice site is formed, resulting in exonization of a portion of the LINE-1 and intron 25 and likely a non-functional transcript. It has been reported that LINE-1 have multiple splice donor and acceptor sites supporting this hypothesis [14]. Preliminary analysis of the RNAseq reads supports the hypothesis of exon-skipping without significant exonization of the intron due to a cryptic splice site, as intron 25 sequences preceding exon 26 or exon 27 sequences are a small minority of all exon 26 or exon 27 sequences. Future experiments should focus on quantification of the contribution of each exon-skipping mechanism to the Tg transcript load.

RT-PCR analysis showed that in heterozygotes approximately 66% of all Tg transcripts contain exon 26, whereas in homozygotes approximately 25% of Tg transcripts contain exon 26. These data indicate that on the LINE-1 insertion allele, exon 26-skipping is not fully penetrant. This results in a mix of mutant and wild type TG protein being translated, albeit more skewed towards mutant in the homozygotes than in the heterozygotes. The presence of wild type TG in the homozygotes could explain the relatively mild phenotype as compared with other TG mutant rodent models.

Several lines of evidence from our work support the hypothesis that the omission of 64 amino acids encoded by exon 26 of the Tg gene promotes protein misfolding and induction of an ER-stress response. Molecular structural analysis revealed an average RMSF of 1.6 Å for the wild type and 2.1 Å for the mutant resulting in greater structural instability despite retention of the overall fold structure. This analysis further illustrated that the tightly packed core in the TG arm region is replaced by large cavities in mutant proteins likely disrupting interactions within the network structure. The modeling is supported by the IHC staining with an anti-thyroglobulin antibody showing basolateral amphophilic accumulation within the ER (confirmed by transmission electron microscopy). The molecular pathway analysis from the transcriptional data provides additional support to the hypothesis of misfolded TG accumulating in the ER leading to a stress response. The most significant hits are known regulators of ER stress-induced inflammatory response [43]. For example, calreticulin, an ER resident protein together with calnexin (an ER integral membrane chaperone) and ERp57 (a PDI-like protein resident in the ER), are responsible for quality control and folding in newly-synthesized (glyco)proteins [44,45]. Further, the glycosylation of newly-synthesized proteins acts as a signal for enhanced folding and quality control and/or for the degradation of misfolded proteins [46]. Similarly, ER-degradation enhancing α-mannosidase-like (EDEM1) is up-regulated in mutant TG and functions to extract terminally misfolded proteins from the calnexin folding cycle targeting them for ER-associated degradation (ERAD) [47]. Finally, several heat shock proteins activated in response to environmental stress are modulated in our analysis. This family of proteins are critical to maintaining cellular homeostasis, namely HSP90, is involved in multiple cellular processes including correcting misfolded proteins in the ER [48,49]. Small HSPs such as HSPA5 and HSPA9 are up-regulated rapidly in response to protein unfolding pressure [48].

There are no published human analogs of this mutation in the *TG* gene despite many well documented instances of diseases resulting from transposable element insertions [50], numerous mutations resulting in thyroid disease and cancer [51,52], and numerous *TG* mutations causing thyroid disfunction [53]. One reason for this may be that humans have evolved more effective "defense" mechanisms to suppress TE mobilization and/or the relatively rapid generation time in mice allowing more opportunities for LINE-1 activation and random insertional events [54,55]. Another reason might be that the human genome has only ~100 full-length, active LINE-1, while the mouse genome has~3,000. Therefore, the chance that a LINE-1 mobilizes and lands in any gene is much higher in the mouse genome than in the human genome. The relatively subtle physiological effect of the intronic LINE-1 identified in this study can only be recapitulated in human if a similar mutation arises in the human intron 25 causing exon 26-skipping with incomplete penetrance, which to our knowledge has not been identified yet.

In conclusion, we describe the identification and initial characterization of a spontaneous mutation underlying thyroid pathology found in the B6NTac strain. Because the mutation was not fixed in the B6NTac strain, it was removed from all colonies through selective breeding. The *Tg^{tdys-Tac}* allele is a novel model for congenital thyroid dysplasia caused by exon 26-skipping due to the presence of an intronic LINE-1.

## Materials and methods

### Animals

All mice were kept in an Association for Assessment and Accreditation of Laboratory Animal Care (AAALAC)-accredited facility at Taconic Biosciences. All procedures were approved by Taconic Biosciences' Institutional Animal Care and Use Committee (IACUC). All studies were performed in compliance with the National Institutes of Health (NIH) Guide for the Care and Use of Laboratory Animals and the Animal Welfare Act. Mice were fed the NIH 31M Rodent Diet. All mice for this study were derived from the commercial C57BL/6NTac breeding colony and maintained by keeping accurate pedigree records.

All animal procedures conducted at Merck & Co., Inc., Rahway, NJ, USA were approved by the Merck & Co., Inc., Rahway, NJ, USA Institutional Animal Care and Use Committee and conducted in an Association for Assessment and Accreditation of Laboratory Animal Care International–accredited facility in compliance with the National Institutes of Health (NIH) Guide for the Care and Use of Laboratory Animals and the Animal Welfare Act.

Euthanasia was performed by $CO_2$ asphyxiation, followed by cervical dislocation, exsanguination via vena cava or cardiac removal following cardiac puncture (for blood serum collection) as a secondary method. This study did not utilize any procedures that may cause more than momentary pain or distress requiring analgesia. At both facilities, standard operating procedures are in place to alleviate animal suffering.

### Histology and immuno-histochemistry and ultrastructure

**Light microscopic evaluation.** Thyroid glands and pituitary glands were collected from C57BL/6 strain mice, fixed in 10% NBF and processed to paraffin blocks. Animals were obtained from Taconic (B6NTac) and Charles River (B6NCrl). Histologic tissues sections 5 µm thick from paraffin blocks containing thyroid glands were mounted on glass slides and stained using either hematoxylin and eosin or for immunohistochemistry utilizing the Leica BOND RX Automated Research Stainer. Tissue sections for IHC were mounted on plus slides and applied citrate pH6 heat-induced epitope retrieval (HIER). Select mouse thyroids from each vendor were immunohistochemically stained using rabbit anti-thyroglobulin (Abcam ab156008) primary antibody at a dilution of 1:20,000. Anti-thyroglobulin primary antibody binding to thyroid tissue sections were chromogenically visualized using BOND ready-to-use polymer refine detection kit (DAB). Concurrent negative control was achieved by omitting the primary antibody and replacing with rabbit IgG at equivalent concentration. Sections of pituitary gland were stained using the same procedures, but for IHC substituting rabbit anti-TSH-B (Invitrogen PA5–32617) dilution 1:200 for the primary antibody.

**Ultrastructural evaluation.** The thyroid gland for ultrastructural evaluations were collected by direct placement of the right lobe into 4% paraformaldehyde and 1% glutaraldehyde in 0.1 M phosphate buffer (4F:1G). Approximately, 1 mm³ thyroid specimens were post-fixed in 2% osmium tetroxide, then processed and infiltrated with epoxy resin on a Lynx II (Electron Microscopy Sciences) automated tissue processor. Specimens were subsequently embedded in LX-112 epoxy resin (LADD Research Industries) and ultrathin sections were obtained on a Leica EM UC6 ultramicrotome. The sections were stained with 2% uranyl acetate and Reynold's lead citrate and examined on a 120 kV FEI Tecnai Spirit Biotwin transmission electron microscope (Thermo Fisher Scientific, Inc.). Images were obtained using the Gatan Orius SC1000 Digital CCD camera (Gatan, Inc.) integrated with the transmission electron microscope.

## Serum hormone level measurements and statistical analysis

Whole blood was collected into Microtainer tubes (BD Biosciences) by cardiac puncture, immediately following euthanasia by $CO_2$ asphyxiation. Serum was extracted by centrifugation as recommended by the manufacturer and transfer of the upper layer to a fresh tube. The serum was immediately frozen and stored at −80°C. Serum T3 and T4 levels were determined using the DRI Thyroxine assay at IDEXX Laboratories. Levels were imported into GraphPad (Prism) software and groups were compared using one-way ANOVA, followed by the Bonferroni correction for multiple comparisons. Significance was determined at an adjusted p-value of 0.05 or less. To estimate the required sample size for the thyroid hormone measurements, we conservatively estimated the effect sizes, based on another mouse model of hypothyroidism due to $Tg$ mutations [28]. For T3, with an effect size of 0.67, standard deviation of 16, difference between the means of 35 and a sample size of minimally 7 per group, the study had 88% (or higher for greater sample sizes) power to reject the null hypothesis at a Bonferroni adjusted p-value of 0.05. For T4, with an effect size of 0.67, standard deviation of 0.8, difference between the means of 1.9, and a sample size of minimally 7 per group, the study had 81% (or higher for larger sample sizes) power to reject the null hypothesis at a Bonferroni adjusted p-value of 0.05. All raw data and statistical analysis have been included in S2 File.

## Genomic and transcriptomic analysis

**Transcriptional analysis of *Tg* in mouse thyroids.** *Tg* mRNA amplification (~8500 bp) was conducted to identify mutations as there are known genetic mutations associated with phenotypic disorders in mouse and human [8,56]. RT-PCR primers were designed spanning overlapping 2 kb regions of the full-length *Tg* transcript to ensure full coverage. Areas that did not amplify successfully were re-evaluated with RT-PCR primers spanning smaller 1 kb regions. To confirm PCR amplification products the primer-probe assays were validated using thyroid mRNA from C57BL/6 mice from an independent vendor (Charles River Labs, CRL). Transcriptional analysis was performed for thyroids of 5 affected and 5 unaffected C57BL/6 mice from Taconic and 5 from CRL. Total RNA was isolated from thyroid with Applied Biosystems' MagMax 96 instrument using the MagMax RNA 96 for microarrays kit (PN1839-4). First strand cDNA was generated using Applied Biosystems' (ABI) High-Capacity cDNA reverse transcription kit (PN43678813). Taq DNA High Fidelity Polymerase was used to perform PCR. Amplification products were visualized using agarose gel electrophoresis.

To evaluate the variant *Tg* transcript level (with and without exon 26), expression analysis by quantitative PCR was performed. Applied Biosystems quantitative PCR assays were selected for exon/exon junctions 24/25 (Mm00447523_m1), 25/26 (Mm00447524_m1), and 26,27 (Mm00447525_m1) of the mouse *Tg* mRNA (RefSeq NM_009375.2) and run according to manufacturer's recommendations. Variant *Tg* transcript levels were calculated as the ratio of the Ct value for the exon 25–26 and exon 26–27 junctions over the Ct value for the exon 24–25 junction, relative to WT. Normalized (to WT) values were imported into GraphPad (Prism) software and groups were compared using unpaired t-tests.

**RNA sequencing.** RNA Sequencing was performed on five affected C57BL/6 mice thyroids from Taconic, one KO unaffected animal from Taconic, and six unaffected from Charles River Labs, to provide an mRNA expression-based

measurement of responses. The Roche KAPA RNA HyperPrep kit with RiboErase (HMR) KR1351 – v1.16 was used for RNA-Seq library construction from 500 ng of total RNA. Sequencing was performed on the Illumina NextSeq 500 platform using the 75-cycle flow cell. Genome alignment and gene quantitation were performed using OmicSoft Array Studio. Reads were aligned to the Mouse.B39 (Ensembl.R96) genome reference using the OmicSoft Aligner (OSA4). The Ingenuity Pathway Application (version 107193442) was used for pathway analysis on the RNA sequencing data set. The raw RNAseq reads have been uploaded to the Gene Expression Omnibus under accession number GSE298337.

**Whole genome sequencing.** DNA from five affected and seven unaffected mice livers was subjected to sequencing on an Illumina NextSeq 2000 instrument, employing the Roche KAPA HyperPlus Kit (KR1145 - v8.21), resulting in 2x155nt paired-end reads. For reads derived from exon 25 of *Tg*, a custom python script was used to count the 10 nucleotides found downstream of the 3' end of exon 25. A custom pipeline was used to detect variants. Analysis methods included not removing PCR duplicates, analyzing all sequenced bases with quality ≥ 20, minimum read coverage was 5, minimum allele fraction was 0.1.

**Long-read targeted nanopore sequencing.** High molecular weight genomic DNA from a homozygous animal was extracted using the Monarch HMW DNA Extraction Kit (NEB). Long-read targeted sequencing was done as described in Gilpatrick et al. (2020) [57] using the Cas9 Sequencing Kit (Oxford Nanopore). The single guide RNA (sgRNA) used to target the LINE-containing intron in *Tg* were TGAAAGGTAGATCCTCAACCAGG, TATCTGGTGTAATAAGATGGGGG, AGCCACTGAGAATGATTGAGGGG, ATATTGGGAAAAACACACCTTGG. Reads were generated using a MinION flow cell on a nanopore sequencer (Oxford Nanopore) and aligned to the mm10 mouse genome using minimap2 [58]. Location-specific reads were extracted from the alignment file based on chromosomal position of the sgRNA using samtools [59]. Reads were assembled using shasta with default settings allowing a minimal read length of 1,000 [60]. The sequence assembly is provided in S1 File.

**Genotyping.** Ear pinnae biopsies were collected and processed to extract genomic DNA using a DNEasy kit (Qiagen). An assay was developed for genotyping at the Taconic Molecular Analysis Lab. The following primers were used to amplify the wild type or mutant allele, respectively: WT_F 5'-TGGTGCATGCCCATGAAGAT-3', MUT_F 5'-GCACCCTCTCACCTGTTCAG-3', R 5'-GGCAAGCAGACAGTAGGTGT-3' using HotStarTaq (Qiagen), and the reaction was run using the following cycling conditions: 95C 15:00'; 95C 0:30', 60C 1:00', 72C 0:45' 35x; 72C 5:00'. Following PCR amplification, the reaction mixture was analyzed using LabChip with DNA 5K reagent kits (Revvity). Amplicon identification was performed using LabChip GX Reviewer software (Revvity).

## Molecular modeling of TG

The structure prediction of the mouse TG Arm region (amino acids 1440-2184) with or without the exon 26 deletion (amino acids 1681-1744) was performed through AlphaFold 2 (AF2) monomer implemented in-house on high-performance computing (HPC) clusters [22–24], and the top-ranked models were used. To examine whether the Arm fragment could be modeled alone, the Arm sequence of human TG was also modeled by AF2, and the output structure closely resembles cryo-EM structure of human TG Arm (fetched from PDB ID: 6SCJ) [22]. Residue-specific pLDDT scores were extracted from output structural models and plotted to compare position-specific prediction confidence and inferred structural disorder. Molecular dynamics simulations were performed on in-house HPC clusters by OpenMM (version 7.7.0) with GPU acceleration (NVIDIA A100 or V100, CUDA Toolkit version 10.2.89) [61,62]. Using AF2-predicted WT and mutant mouse TG Arm fragment, 125 amino acids to the N terminus and 209 amino acids to the C terminus of the exon 26 peptide, corresponding to two domains in direct contact with the excised sequence, were included for simulation with or without the exon 26 peptide. Input structures were first standardized using the "pdbfixer" module of OpenMM, and then prepared under Amber 14 forcefield with Generalized Born (gbn2) implicit solvent, 2 nm non-bonded cutoff, and H bond length constraints. Positional restraints were also placed on Cα atoms of 15 amino acids at the N terminus and 9 amino acids at the C terminus, which correspond to the first secondary structure elements at the two ends (S4 Fig), at 100 kJ·mol⁻¹nm⁻¹.

Simulations were performed using Langevin integrator at $1\,ps^{-1}$ friction coefficient and 2 fs step size. Structures were first energy minimized, and then equilibrated to 300 K in 100 steps before 500 ns of production run. Snapshots were taken every 0.5 ns during production run and stored for trajectory analysis. Five repeated simulations were performed for each structure. Output trajectories were analyzed by CPPTRAJ (Amber 20) on backbone atoms C, CA, N, and O in respect to the first frame to report RMSD and residue-specific RMSF, which were then averaged across all five repeats. CPPTRAJ was also used to calculate average structure for each simulation. Amber topology files for input structures were prepared by MOE (2022.02, CCG) to be used for CPPTRAJ. Cavity analysis was performed on PyMOL (2.3.1, Schrodinger) [63].

## Supporting information

**S1 Fig.** *Tg* **RT-PCR from Thyroid RNA. (A)** Schematic representation of the full-length *Tg* mRNA (solid line) displaying the overlapping amplicons (dashed arrows). **(B)** Gel electrophoresis images showing amplification results of the amplicon covering exon 17–22 (upper panels) and amplicon covering exon 22–32 (lower panels).
(TIF)

**S2 Fig. Time course thyroid phenotype progression by genotype.** Shown are representative microscopic images of H&E-stained thyroid follicular epithelial cells from B6NTac animals WT, HET, or HOM for the LINE-1 in *Tg* at 4 wks, 6 wks, 12 wks, 9 months, and 12 months of age.
(TIF)

**S3 Fig. Body weight (BW) measurements.** BW measurements of B6NTac animals WT, HET and HOM for the LINE-1 in *Tg* at 4 wks, 6 wks, 12 wks, 9 mo and 1 yr of age, split by sex. The data were analyzed by One-way ANOVA with Bonferroni correction. Significance is indicated by * (adj. $p < 0.05$).
(TIF)

**S4 Fig. Representative average structure of MD simulation trajectories during production run for WT and mutant mouse TG models.** The 15 N-terminal and 9 C-terminal segments with positional restraints placed during simulation were colored as red.
(TIF)

**S1 File. Consensus sequence assembly of the LINE-1 in intron 25 of mouse** *Tg***, derived from long-read sequencing.**
(TXT)

**S2 File.** Raw data and statistical analysis for Figs 3, 5 and S3.
(XLSX)

## Acknowledgments

The authors would like to thank Tara McNutt for conducting the IHC work and Kristen Flor for performing the TEM work.

## Author contributions

**Conceptualization:** Wendy J. Bailey, Bart M. G. Smits, Thomas W. Rosahl, Thomas Forest.

**Data curation:** Bart M. G. Smits, Pamela Lane, Sabu Kuruvilla, Melissa MacGowan.

**Formal analysis:** Bart M. G. Smits, Zoltan Erdos, John M. Gaspar, Pamela Lane, Sabu Kuruvilla, Douglas Thudium, Jingzhou Wang, Warren E. Glaab, Heather Multari, Christine Cumo, Adam Navis, Thomas Forest.

**Methodology:** Zoltan Erdos, John M. Gaspar, Pamela Lane, Sabu Kuruvilla, Douglas Thudium, Jingzhou Wang, Warren E. Glaab, Melissa MacGowan, Heather Multari, Christine Cumo, Adam Navis, Thomas Forest.

**Project administration:** Wendy J. Bailey, Bart M. G. Smits, Thomas W. Rosahl, Melissa MacGowan, Thomas Forest.

**Resources:** Wendy J. Bailey, Adam Navis.

**Supervision:** Wendy J. Bailey, Bart M. G. Smits, Thomas W. Rosahl, Thomas Forest.

**Visualization:** Wendy J. Bailey, Bart M. G. Smits, John M. Gaspar, Jingzhou Wang, Adam Navis, Thomas Forest.

**Writing – original draft:** Wendy J. Bailey, Bart M. G. Smits, Zoltan Erdos, John M. Gaspar, Thomas W. Rosahl, Jingzhou Wang, Thomas Forest.

**Writing – review & editing:** Wendy J. Bailey, Bart M. G. Smits.

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
