## [Decision Letter · Decision Letter 0]

18 Feb 2025

PONE-D-24-47670LINE-1 Transposition into Murine Thyroglobulin Results in Congenital Thyroid DysplasiaPLOS ONE

Dear Dr. Smits,

Thank you for submitting your manuscript to PLOS ONE. After careful consideration, we feel that it has merit but does not fully meet PLOS ONE’s publication criteria as it currently stands. Therefore, we invite you to submit a revised version of the manuscript that addresses the points raised during the review process.

We look forward to receiving your revised manuscript.

Kind regards,

SARBASHRI BANK, PhD

Academic Editor

PLOS ONE

Journal Requirements:

2. To comply with PLOS ONE submissions requirements, in your Methods section, please provide additional information regarding the experiments involving animals and ensure you have included details on methods of anesthesia and/or analgesia, and  efforts to alleviate suffering.

3. We note that your Data Availability Statement is currently as follows: “All relevant data are within the manuscript and in Supporting Information files.”

Additional Editor Comments:

This work offers an intriguing look at the relationship between thyroid dysfunction and retrotransposon insertion.

The impact of retrotransposon insertions on gene regulation, which results in physiological problems, is highlighted in this paper. It is a good model for investigating thyroid disease in mice because in this instance, the LINE-1 insertion results in thyroid dysplasia and hypothyroidism, which then develops to thyroid adenoma. This sheds light on how mutations caused by retrotransposons can have serious effects on health and illness, particularly in commonly used inbred mouse strains.

But have few questions:

1) How line-1 insertion affect splicing mechanism in the context of cis-acting regulation or spliceosome have not been explored

2) Effect of UPR on metabolic dysregulation, apoptosis and cell stress are explored

3) Authors also did not show how mutation mutation affect other gene or signalling pathway

4) The mutation is stably inherited across generation?

5) Authors also didn’t mention any therapeutic intervention.

Reviewers' comments:

Reviewer's Responses to Questions

Comments to the Author

1. Is the manuscript technically sound, and do the data support the conclusions?

Reviewer #1: Yes

2. Has the statistical analysis been performed appropriately and rigorously? 

Reviewer #1: Yes

3. Have the authors made all data underlying the findings in their manuscript fully available?

Reviewer #1: Yes

4. Is the manuscript presented in an intelligible fashion and written in standard English?

Reviewer #1: Yes

5. Review Comments to the Author

Reviewer #1: According to my understanding, the study investigates a spontaneous mutation in the C57BL/6NTac mouse strain caused by the insertion of a LINE-1 retrotransposon into intron 25 of the thyroglobulin (Tg) gene. This mutation disrupts normal splicing, resulting in the skipping of exon 26 and the production of a variant TG protein lacking 64 amino acids critical for structural stability. Affected mice have congenital thyroid dysplasia, induced by ER stress and the accumulation of a misfolded TG protein, and develop thyroid adenomas by 9 months of age in homozygous mice. Serum analyses show decreased T3 and T4 levels with increased TSH levels, hence indicating hypothyroidism. It's inherited in an autosomal dominant way, and homozygous mice show more severe phenotypes than heterozygous ones. Molecular modeling shows that the deletion destabilizes the TG protein, leading to misfolding and cellular stress. This work provides a new model for congenital thyroid dysplasia and highlights the potential role of retrotransposons like LINE-1 in disrupting gene function and causing disease.

The manuscript seems technically sound, and it has robust methodology integrating genomic, transcriptomic, molecular modeling, and histological analyses. Identification of a LINE-1 retrotransposon insertion leading to exon skipping in the Tg gene is well-supported by whole-genome sequencing and RNA-seq data demonstrating the molecular basis for aberrant splicing. Moreover, the functional consequences of this mutation, such as protein misfolding, ER stress, and thyroid dysplasia, are convincingly demonstrated by molecular modeling, immunohistochemistry, and hormonal assays. However, although the data generally support the conclusions, some aspects, such as the precise molecular mechanism of exon skipping and its broader implications for human thyroid disease, are less extensively explored. Allowing and providing details such as effect sizes and confidence intervals in statistical analyses while addressing the gaps in the data will strengthen the link even further between the data and the conclusions.

However, I have some pertinent questions while reviewing the manuscript.

1. Could you elaborate on how the LINE-1 insertion specifically disrupts splicing at the molecular level? Are there cryptic splice sites involved?

2. Are there any known human analogs of this mutation? How does this model reflect human thyroid dysplasia mechanisms?

3. What factors might account for the variability in phenotype severity between heterozygous and homozygous mice? 4. Based on your findings, are there specific therapeutic strategies you foresee for conditions involving LINE-1-induced dysregulation?

5. Have you considered exploring compensatory mechanisms in heterozygous carriers that prevent progression to adenoma?

The statistical analysis in the paper is moderately clear and appropriate for validating key findings, but there are areas of improvement. The study makes use of unpaired t-tests in comparing serum hormone levels, T3, T4, and TSH across genotypes, and molecular dynamics metrics like RMSF for assessing protein stability, which are suitable for their respective analyses. Results are described with proper p values (e.g., (p = 0.02, p < 0.01), providing a precise foundation for assertions. The work is not transparent about other essentials, including how power analyses explain sample sizes for each experiment or confidence intervals plus effect sizes, which would help further contextualize their findings. While sample sizes sometimes vary between experimental conditions, even more importantly and unfortunately, all experiments lacked multiplicity adjustments. The biological import of statistically significantly different values in hormones, such as hormone concentration levels, should also be appreciated better. Overlayers, such as raw data or error bars, onto graphical representations are useful for highlighting data distribution better. These changes will strengthen analysis through more specified statistical methodologies about the study adjusted for multiple comparison tests and providing clearer data illustrations.

The authors have declared that all data underlying the findings in their manuscript are fully available without restrictions. They specify that the relevant data are included within the manuscript and its supporting information files, ensuring accessibility for further analysis or validation by the scientific community.

Overall, authors have put a lot of effort into this research. The methodological robustness of the study is remarkable, which, by adopting an all-round approach integrating genomic, transcriptomic, molecular modeling, and histological techniques, provides the study with the strength and reliability of its findings. Besides, the study has great significance since it not only contributes to the basic scientific knowledge of retrotransposons but also contributes to their role in gene expression, providing a translational approach toward understanding and treating thyroid diseases.

I have a few suggestions for the strengthening of the rationale of research. Although the study manages to point out exon skipping as the critical event giving rise to the observed thyroid dysplasia, it is unclear precisely what the molecular mechanism could be by which the LINE-1 insertion impacts splicing. Further experimentation to discern whether cryptic splice sites or other mechanisms are involved would deepen mechanistic understanding. In addition, more comparison to other Tg-related mutations in mice or humans would give context and emphasize this model's uniqueness and limitations. Lastly, the broader implications of the findings could be expanded on, especially how this model might inform understanding of human thyroid disease and other conditions that may be caused by LINE-1 or similar retrotransposons.

The study offers valuable insights into the structural effects of exon 26 deletion in the TG protein through molecular dynamics (MD) simulations. Nevertheless, there are several areas that could be enhanced in the analysis. First, the focus is on the arm fragment rather than the full-length protein, thereby limiting the context of the simulation. The entire TG protein should be simulated in future studies to capture the more comprehensive structural and functional implications of the mutation. Validation of the computational results through experimental techniques such as cryo-EM or NMR would add more credibility to the findings. The study only reports RMSF for assessing flexibility but would benefit by including RMSD analysis as well. RMSD would offer more accurate results of general structural stability and corresponding conformational shifts, making the conclusions drawn in terms of the mutation's impact on TG stability stronger.

It would be great if this could be conducted to explore how the observed structural changes translate to functional consequences for TG, including their potential impact on thyroid hormone synthesis. Further interaction of mutant TG with cellular factors like chaperones is a way to get a more comprehensive understanding of the effects that the mutation could have.

6. PLOS authors have the option to publish the peer review history of their article (what does this mean? ). If published, this will include your full peer review and any attached files.

**Do you want your identity to be public for this peer review?** For information about this choice, including consent withdrawal, please see our Privacy Policy .

Reviewer #1: **Yes: ** SAMUDRA PAL

---

## [Author Response · Author response to Decision Letter 1]

11 Apr 2025

The response to the editor and reviewer has been uploaded in a file named Response to Reviewers.

---

## [Decision Letter · Decision Letter 1]

14 May 2025

LINE-1 Transposition into Murine Thyroglobulin Results in Congenital Thyroid Dysplasia

PONE-D-24-47670R1

Dear Dr. Smits,

We’re pleased to inform you that your manuscript has been judged scientifically suitable for publication and will be formally accepted for publication once it meets all outstanding technical requirements.

Kind regards,

SARBASHRI BANK, PhD

Academic Editor

PLOS ONE

Additional Editor Comments (optional):

Authors responded sufficiently to all the comments. But before final acceptance, authors should correct minor changes as told by reviewer-1.

Reviewers' comments:

Reviewer's Responses to Questions

**Comments to the Author**

1. If the authors have adequately addressed your comments raised in a previous round of review and you feel that this manuscript is now acceptable for publication, you may indicate that here to bypass the “Comments to the Author” section, enter your conflict of interest statement in the “Confidential to Editor” section, and submit your "Accept" recommendation.

Reviewer #1: All comments have been addressed

2. Is the manuscript technically sound, and do the data support the conclusions?

Reviewer #1: Yes

3. Has the statistical analysis been performed appropriately and rigorously? 

Reviewer #1: Yes

4. Have the authors made all data underlying the findings in their manuscript fully available?

Reviewer #1: Yes

5. Is the manuscript presented in an intelligible fashion and written in standard English?

Reviewer #1: Yes

6. Review Comments to the Author

Reviewer #1: Summary

This manuscript investigates the impact of a LINE-1 insertion in exon 26 of the thyroglobulin (TG) gene on splicing, protein stability, and thyroid function using mouse models. The authors characterize both heterozygous and homozygous phenotypes through RNA-seq re-analysis, qRT-PCR, molecular dynamics (MD) simulations, hormone assays, and transmission electron microscopy. Overall, the study is well conceived, employs appropriate methods, and the authors have satisfactorily addressed all reviewer and editor comments in their revised submission.

Major Points

1. Splicing Mechanism

o Comment: Elaborate on how the LINE-1 insertion disrupts splicing.

o Response & Revision: The authors added a discussion of potential branch-site interference and ruled out significant cryptic-site exonization based on re-analysis of RNA-seq reads (Discussion, pp. 17–18).

o Status: Adequately addressed.

2. Human Analogs

o Comment: Are there known human TG LINE-1 insertions?

o Response & Revision: New paragraph clarifies no reported human analogs and speculates on selective barriers (Discussion, p. 19).

o Status: Adequately addressed.

3. Genotype–Phenotype Dosage

o Comment: Account for variability between heterozygotes and homozygotes.

o Response & Revision: Added quantitative RT-PCR data showing 63% vs. 23% exon 26 inclusion in HET vs. HOM and a gene-dosage explanation (Results, p. 9).

o Status: Adequately addressed.

4. Molecular Dynamics Analyses

o Comment: Include RMSD plots; consider full-length TG modeling.

o Response & Revision: RMSD plot added (Fig. 6E; Results, p. 13); full-length modeling deferred to future work.

o Status: Adequately addressed.

5. Compensatory Mechanisms in Heterozygotes

o Comment: Explore compensatory pathways.

o Response & Revision: Discussion reiterates a gene-dosage model and notes no additional pathways detected.

o Status: Adequately addressed.

6. Statistical Methods

o Comment: Clarify power analyses, multiple comparisons, effect sizes, and confidence intervals.

o Response & Revision: Replaced t-tests with one-way ANOVA plus Bonferroni corrections (Methods, p. 23); added power calculations for hormone assays.

o Status: Adequately addressed.

7. Editor’s Requests

o Supporting Information captions moved to end of manuscript; in-text citations updated.

o Ethics and anaesthesia details added (“CO₂ asphyxiation, followed by cervical dislocation…” in Methods).

o Data availability: Raw data in S2 File with a “GEOxxxx” placeholder.

o UPR pathway discussion and TEM confirmation remain integrated.

o Status: Adequately addressed, pending GEO accession finalization.

Minor Suggestions Before Final Acceptance

1. GEO Accession: Replace “GEOxxxx” placeholder with the actual accession number once available.

2. Figure Legends: Explicitly state “one-way ANOVA with Bonferroni correction” in legends reporting adjusted p-values.

3. Effect‐Size Reporting: Consider including confidence intervals or η² values for ANOVA results to contextualize biological significance.

4. Mechanistic Validation: If feasible, add targeted RT-PCR assays probing for any rare cryptic splice junctions to strengthen the splicing‐interference hypothesis.

Recommendation:

Accept pending minor revisions as outlined above.

7. PLOS authors have the option to publish the peer review history of their article (what does this mean? ). If published, this will include your full peer review and any attached files.

**Do you want your identity to be public for this peer review?** For information about this choice, including consent withdrawal, please see our Privacy Policy .

Reviewer #1: No

---

## [Editor Report · Acceptance letter]

PONE-D-24-47670R1

PLOS ONE

Dear Dr. Smits,

I'm pleased to inform you that your manuscript has been deemed suitable for publication in PLOS ONE. Congratulations! Your manuscript is now being handed over to our production team.

Kind regards,

on behalf of

Dr SARBASHRI BANK

Academic Editor

PLOS ONE